# Learning by Doing and Training Satisfaction: An Evaluation by Health Care Professionals

**DOI:** 10.3390/ijerph16081397

**Published:** 2019-04-18

**Authors:** Marta Gil-Lacruz, María Luisa Gracia-Pérez, Ana Isabel Gil-Lacruz

**Affiliations:** 1Health Science Faculty, University of Zaragoza, 50009 Zaragoza, Spain; 2Social Sciences and Work Faculty, University of Zaragoza, 50009 Zaragoza, Spain; mlgracia@unizar.es; 3School of Engineering and Architecture, University of Zaragoza, 50018 Zaragoza, Spain; anagil@unizar.es

**Keywords:** training satisfaction, context-based learning, Learning by Doing, course evaluation, impact assessment

## Abstract

As one of the training methodologies employed in the health care context, ‘Learning by Doing’ prioritizes the transference of competence, control, and workplace motivation. However, there are few published works that consider the opinions of health care professionals in relation to the effects of this kind of training on their workplace competence. The goal of this research was to evaluate the level of satisfaction and impact on quality of care, as perceived by the health care professionals that participated in this training program (Formative Focus). The evaluation utilized an online questionnaire that assessed professional satisfaction through five dimensions: The training methodology; the training program; its economic impact; perceived improvement in professional competence; and, perceived improvement in quality of care. A total of 364 health professionals took part in the training course and were asked to complete an online questionnaire. The variables that contemplated satisfaction were related to quality of care and preferences regarding the training methodology. Participants preferred interactive methods for learning with emphasis on practical contents. In terms of application of learning to their jobs, health care professionals found Learning by Doing skills more useful to transfer to their workplace. This experience of Learning by Doing training indicates an interesting potential for impact on improvement: Quality of health care, health care competence of professionals, and organizational development.

## 1. Introduction

The current model of the clinical administration of the health service prioritizes the principles of integrated patient treatment, efficient management of organizations, and good governance based on transparency and social responsibility [1]. The health organization is expected to adopt strategies for the improvement of the health system that are oriented to total quality management [2].

Donabedian [3] suggests that quality of health care is determined by a complex combination of technical and scientific issues, interpersonal relationships, and other elements of the health care environment. The participation of the implicated agents is essential for improving quality. The scientific literature evidences the importance of considering patient satisfaction with the results and processes of their treatment, the quality of service, and the work of the health professionals. Patient satisfaction can be measured by objective questions on the efficacy of the use of health resources, for example treatment adherence and preventative measures [4,5].

The participation of the health professionals is also a determinant in the evaluation of quality of care. The level of quality of the services offered by a health system is directly related to the degree of satisfaction of the professionals that work in that system [6].

For Robbins [7], job satisfaction can be defined as ‘a positive feeling about one’s own job resulting from an evaluation of its characteristics’ (2004 p.79). A high level of job satisfaction means that the employee will have positive feelings towards their work, and vice-versa. Based on this premise, job satisfaction is directly related to the attitudes of the professional and their conditions of work.

Güleryüz, Güney, Aydın, and Aşan [8] have divided the factors that influence job satisfaction into two groups: Intrinsic and extrinsic. Intrinsic factors include personal development (possibilities for training and promotion), degree of recognition, responsibility and autonomy, work activity, and work objectives. Extrinsic factors include salary, relationships with supervisors and colleagues, working conditions, status, and employment security.

In the health care services, a promising research line has linked job satisfaction to the quality of working life and work environment. Quality of working life (QWL) is a complex concept that contemplates working hours, salary, the workplace environment, benefits and services, possibilities for professional promotion and improvements, access to information, human relationships, and other factors that are relevant to job satisfaction and performance [9].

The absence of satisfaction among health service professionals (in general, or with a specific element: Job security, personal relationships, opportunities for promotion etc.) has been shown to lead to stress or burnout [10]. Determinants of stress among Spanish health professionals include excessive workload (56.3%), carrying out routine tasks (14.8%), lack of recognition (9.1%), and lack of continuous training (8.3%) [11]. A study on primary care reported similar causes for stress: Excessive pressure, excessive responsibility for decisions above the primary care level, and lack of access to continuous training [12].

Two of the negative effects of stress are that workers do not identify with the health organization and that their performance decreases [13]. Interventions are necessary at a number of levels and training is a vital component. Workplace training can be an occupational incentive and a basic motivational measure; training can allow for the development of the potential of the workers and be a cornerstone of an employee rewards strategy [14]. Professional training is further associated with greater occupational autonomy, professional recognition, and promotion [15].

Along with job security and participation, training has been shown to be significant with regards to improving health service workers’ evaluation of available organizational resources [16]. In a study of primary care professionals undertaken by Martín, Fernández, Gómez, and Martínez [11], 6.9% of participants called for increased investment in training.

Professional training can therefore be understood as both a positive measure for improving the quality of the system and a strategy for the prevention of stress and an improved working environment. Studies on ‘Magnet Hospitals’ (hospitals that the American Nurses Credentialing Center recognizes as those that attract and retain personnel) have indicated that they share a number of characteristics such as professional autonomy and responsibility, leadership, continuous training, and meaningful research programs [17].

Training as an element that impacts on motivation and job satisfaction is also related to the constant development of medical knowledge and technology; it is clearly of vital importance that hospitals and medical centers keep their staff up-to-date on the latest scientific advances [18]. Training programs can improve psychological wellbeing by increasing the health professionals’ control over their work and fostering social support by their colleagues and managers [19].

Continuous training allows the professional to upgrade their knowledge and skills, improving self-confidence and motivation in order to develop a guaranteed quality of care [20]. Training can further be considered as a strategy for optimizing professional activity which can lead to promotion and professional growth which, in turn, generates motivation [21].

In the context of the health service, continuous training can be a motor for organizational change, for two reasons: (i) It positively influences job satisfaction; and, (ii) it is part of a process of improving professional competence that concerns [22] ‘doing’ (acquiring knowledge that the professional needs for the efficient performance of their activities), ‘knowing’ (the analysis and interpretation of ‘doing’), and ‘demonstrating’ (the utilization of what has been learnt). In other words, professional satisfaction is not just related to the acquisition of knowledge and skills, it also encompasses the application of that knowledge and skills and the improvement in competence and quality of care, as perceived by the professional [23]. In the implementation of a training program, the selection of methodology is fundamental to success and achieving objectives [24].

‘Learning by Doing’ is one of the training methodologies most relevant to the health care environment. It is a form of direct instruction in which practical training is given whilst the person is working [25]. Scientific evidence has shown that through this methodology the participant learns quickly and well, and results are only surpassed by individual instruction and are considerably superior to traditional teaching practices [26].

Miller [22] believes that professional training must be applicable to the workplace; Learning by Doing does not absolutely guarantee the fulfilment of this condition but it does accord with the theoretical and methodological foundations established by Dewey [27], Kolb [28], and Borgnakke [29]. It is a system that offers a guarantee of quality because instruction takes place in the organizational context [26].

The utility of this model was demonstrated in a study of resident doctors [30]; a considerable part of their academic clinical practice training takes place in the working environment of a hospital. However, despite the fact that much of the academic training of doctors and nurses is practical and direct, many of the continuous training programs of the health service are based on traditional teaching and learning methodologies [31].

The objective of this study was to determine if a Learning by Doing training course had a positive influence on the professional competence, quality of care, and the job satisfaction of the participants.

## 2. Materials and Methods

### 2.1. Design

An evaluation model was designed to measure the impact of a training program using variables that were defined as outcomes (professional competence, quality of care, etc.) and the final product (professional satisfaction). The study was multi-variable and cross-sectional, based on the results of an online questionnaire completed by course participants.

The study took place in 2014; participants comprised health service professionals working in the Aragon Health Service (Spain). The training program, known as Formative Focus, is aimed at improving diagnostic, surgical and therapeutic skills and is led by the Aragon Institute of Health Sciences. The defining characteristics of this particular course are that the instructors are colleagues that work alongside the participants and that the course takes place in the context of the working environment.

### 2.2. Universe and Sample

A total of 364 doctors and nurses participated. Given the number of subjects, it was decided that it was not necessary to select a smaller sample, so the study universe became, in effect, the final sample. In this way, the 364 health care professionals who joined the training program (Formative Focus 2014) were included in this evaluation process.

There were two professional categories: Physicians and nursing staff. The two groups represented 94.54% of the course participants.

The online questionnaire was sent out four months after completion of the course. Anonymity was guaranteed, and the nature of the research was explained. The questionnaire was to be returned within 10 days of receipt.

A total of 182 valid questionnaires were received; the participation rate was exactly 50%. The fact that the questionnaire was online may have influenced the number of replies. The questionnaires that were received were a reasonably accurate representation of the sociodemographic and professional characteristics of the study universe.

The average age of participants was 44.41 (95% Confidence Interval (CI): 43.17–45.66). A total of 78.8% were women, and 21.2% were men. Replies were received from 77 physicians (42.7% of the sample) and 105 nursing staff (57.3%). The most common areas of work were primary care (103), specialists (54), and emergencies (25). The average number of years that participants had been working in the Aragon Health Service was 17.54.

### 2.3. Instrument

Following the systemic models of the evaluation of training used in previous works [32,33,34], the dimensions and variables of the evaluation instrument were the main dimensions of the organization and the training intervention itself:Satisfaction with the training methodology, comprising two variables: (a) Participants’ opinions concerning the Learning by Doing method, in comparison with other training techniques that they had experienced (e.g., classroom teaching, workshops, conferences, etc.). Response options were 1-‘Yes’ (satisfied) or 0-‘No’ (not satisfied); and, (b) the degree of knowledge/skills retention, as an indicator of being able to utilize the information at work. Results were in accordance with a Likert scale from 0 (none), the lowest retention level, to 3 (a lot), the highest. The dimension combined the above variables and gave values between 0 and 4.Satisfaction with the training program. Target variables were: (a) The instructor; (b) the course content; (c) the didactic approach; (d) time invested in the course; and, (e) program management by the institution. Response options for the five variables were 1-‘Yes’ (satisfied) or 0-‘No’ (not satisfied). The results for the variables were integrated to give an overall value for the dimension between 0 (no satisfaction) and 5 (completely satisfied).Satisfaction with the improvement in professional competence. The following variables [22] were measured by means of a Likert scale from 0 (no satisfaction) to 3 (completely satisfied): (a) Acquisition of new knowledge/skills; (b) usefulness for work; (c) applicability of knowledge/skills; (d) improvement of professional knowledge/skills; and, (e) cascade training (transmission of knowledge and skills to other colleagues. This section of the questionnaire included the option of adding supplementary comments. The overall value for the dimension was a score between 0 and 9 (the higher the score, the more positive the evaluation).Satisfaction with the improvement in quality of care. Participants were asked if they felt that their skills in the areas of diagnosis, treatment, and care had improved after they had taken the course. Scores were 0 (no) or 1 (yes).Satisfaction with the economic repercussions. Two variables were considered: (a) Optimization of recourses—participants were asked if they thought the application of what they had learnt on the course would lead to better use of the resources of the health center where they worked; response options were 0 (no), or 1 (yes); and, (b) impact on expenditure—participants were asked if they thought the course had an impact on the expenditure of their health center or service; response options were 0 (an increase in expenditure), 1 (no impact), or 2 (reduction). The overall value for the two variables was given on a scale of 0–3.

In total, there were 15 items (variables and dimensions) in the questionnaire that concerned an evaluation of the training program. There were also a series of questions referring to the sociodemographic characteristics of the participants (age, sex, professional classification, years working in current job, and number of training activities undertaken in the program). Further information on this instrument and its prior development is available on Gracia-Pérez (2015) [35]. The statistical analysis utilized SPSS v.15 (IBM, Armonk, NY, USA) and Epidat 4.0 (Directorate General of Public Health, Galicia Government, Santiago de Compostela in collaboration with Pan American Health Organization Washington, USA; University of Zaragoza license). 

To carry out the content validity of the questionnaire, the technique ‘Expert Opinion’ was used. As recommended by the technique, it is advisable to consult a number of judges, representative of the different groups in the sample and higher than the number of dimensions to be evaluated (from double to ten times according to Argimón and Jimenez [36]). Hence, to assess the final instrument of 5 dimensions (satisfaction with the training methodology, satisfaction with the training program, satisfaction with the improvement in professional competence, satisfaction with the improvement in quality of care, satisfaction with the economic repercussions), a sample of 10 representative health professionals was selected (5 physicians, 5 nursing staff). This sample revised the questionnaire and gave correct estimations regarding the dimensions, variables, and categories. The Cronbach alpha gave a score of 0.76 for the internal consistency of the items.

This was followed by descriptive and bivariate analyses and a series of predictive models based on univariate binary logistic regressions.

### 2.4. Ethics Approval And Consent to Participate

We obtained ethical approval from the Committee of Research Ethics of the Community of Aragon (CEICA). This Committee is in charge of the evaluation of all research projects with people, human biological samples or personal data, including clinical trials and observational studies with medication. All study participants read the information sheet and agreed to participate in the study voluntarily. They answered the questions and gave written consent.

## 3. Results

### 3.1. Descriptive Results

Satisfaction with the training methodology: (a) All participants (100%) recognized the positive effects of the Learning by Doing methodology compared to other training programs; (b) 70.1% stated they felt that they had improved their knowledge and skills and 20.4% believed they had improved ‘a lot’. The average score (0 to 4) was 3.11 with a standard deviation of 0.54.Satisfaction with the training program: The percentages for satisfaction were very high in all the variables—93.8% for the instructor; 94.4% for course content; 96.2% for didactic approach; 87.1% for time invested in the course; and, 95.1% for program management by the institution. The average score for general satisfaction (0 to 5) was 4.67 with a standard deviation of 0.82.Satisfaction with the improvement in professional competence: (a) Acquisition of new knowledge/skills—59.7% were very satisfied, and 35.2% were quite satisfied; (b) 99.5% of the sample stated that the course was very useful for their work; (c) applicability—only 13.2% said that they could not apply the knowledge/skills they had learnt, mainly due to the lack of patients, equipment, or time; (d) 91.6% believed that the course had led to an improvement of their professional knowledge and skills; and, (e) 79.6% said they were able to transmit their new knowledge and skills to their colleagues. The average score (0 to 9) for the variables related to professional competence was 7.49 with a standard deviation of 1.31.Satisfaction with the improvement in quality of care: 95.7% said that they felt the course had improved their diagnostic, treatment, and care skills at work (scores of 0 = no and 1 = yes).Satisfaction with the economic repercussions: (a) 79% gave a positive response regarding better use of resources; (b) 73.7% believed that the training course would not lead to an increase in health care expenses. The average score (0 to 3) was 1.25 with a standard deviation of 1.11.

### 3.2. Inferential Results

The dependent variables for the statistical analysis were satisfaction with the improvement of knowledge and skills, applicability of knowledge and skills, and improvement in quality of care. To avoid overlapping the inferential statistical techniques, the dimension of professional competence and the variables of improvement and applicability of knowledge and skills were removed and analyzed independently. The interactions between the dimensions and variables are given below.

Satisfaction with the improvement of knowledge and skills was aimed at understanding the main relationships between the characteristics of the participants, their level of satisfaction with the training, and the applicability of what was learnt to their jobs. Chi squared and Fisher tests were undertaken, and results showed significant relationships between all the variables. People that were satisfied with the applicability of the knowledge and skills to their daily work were also satisfied with the training methodology, the improvement in their professional competence, and quality of care.

The same tests were carried out with regards to satisfaction with the improvement in professional competence and improvement in quality of care. Both were significantly related to the other variables, with the exception of the sociodemographic characteristics. It was not possible to contrast hypotheses with the training methodology variable as 100% of the participants stated that they were satisfied.

In addition to the above described bivariate analysis of the variables, the relationships between the main variables and the dimensions were also examined, using the Mann-Whitney U test. The results are shown in Table 1.

Each variable was assigned a scale of values. The p value indicates if there are significant differences between the average values of those that stated that they were not satisfied with the training course, when compared with those that were satisfied. Participants that were satisfied with the training course were also satisfied with the applicability of knowledge gained and with the improvement in quality of care. Figure 1 shows the differences between the averages of those that were satisfied with the course, the improvement in professional competence, the Learning by Doing methodology, and the optimization of resources.

The results of the bivariate analysis confirmed that the three main variables were related with the other variables and the dimensions.

There were statistically significant differences between the three main variables.

Finally, the satisfaction with the improvement in quality of care variable was considered as a final product and analyzed together with satisfaction with the applicability of knowledge and skills and satisfaction with improvement in professional competence.

From the results it could be inferred that:Satisfaction with the improvement in quality of care had a statistically significant relationship with a positive evaluation of the improvement of competence and satisfaction with the applicability of the knowledge and skills that were learnt on the training course.A total of 98% of participants were satisfied with the course and believed that they had improved their professional competence, were able to apply the new knowledge and skills at work, and/or had improved the quality of care. Participants that were not satisfied with the course and did not feel that their professional competence had improved gave a mean score of 2.83 (1.72); this contrasts with those that were satisfied and felt their professional skills had improved who gave a mean score of 4.75 (0.66). Differences were statistically significant (*p* = 0.000).The analysis confirmed a relationship between satisfaction with the applicability of the knowledge and skills learnt on the course and satisfaction with the improvement in quality of care; 92.2% of participants were in agreement and responses were consistent.

Similarly, the relationship between improvement of professional competence and quality of care was also confirmed; 94.4% of participants were in agreement and responses were consistent (Table 2).

Finally, a predictive model for the quality of care variable was formulated using logistic regressions (Table 3) with unified values for four of the five dimensions (none of the variables from the sociodemographic dimension were included as they had almost no impact on the bivariate analysis).

The crude OR (univariate) of all the variables was interpreted as an increment in the probability of the improvement of quality of care when an independent variable was increased. With the adjusted OR (multivariate), the only significant dimension was the training methodology; when its value was raised, there was a significant increase in quality of care. All the other dimensions were constant.

## 4. Discussion

This current research has shown how continuous training can be defined as a variable that has an impact on occupational satisfaction, perceived by professionals as a means to improve their skills, competences, and the quality of their work. In line with the findings of Miller [22], the new knowledge and skills acquired value when they were applied in the workplace and led to individual and organizational improvements that, in turn, increased the professionals’ satisfaction.

This tendency is maintained over time; four months after the training course, descriptive results indicated that 94% of participants believed that they had improved their knowledge to a large or great extent. The statistical significance tests showed that learning was related to the applicability of the new knowledge and skills, the improvement of professional competence, and the improvement in quality of care. However, these data are divergent with the results of other studies about health care environments where knowledge and attitude change due to education programs, but remain only for a short period of time [37,38,39]. Therefore, follow up intervention programs and assessments linked to education and practice require to be planned as a key point of continuous training [40].

In our research, special attention should be paid to the training methodology; it was unanimously agreed that Learning by Doing was the most suitable method for workplace training, as all participants (100%) said that they preferred it to alternatives such as face-to-face classes, seminars, workshops, or conferences. This preference could be explained by the education tradition in health care professions as a form of apprenticeship [41] in which it is important to have direct and active experience, for example by inter-professional simulation programs [42]. In this sense, the program that we have evaluated (Formative Focus) combines the Learning by Doing strategy with inter-professional collaboration. In previous studies, this combination obtained excellent results also in terms of patient safety [43].

The bivariate analysis of the training methodology variable with the variables of applicability of knowledge and skills and improvement in professional competence found no statistically significant differences. The statistical analysis of the sum of the variables indicated that the average score for the participants that preferred the methodology and applied the skills and knowledge in their work was higher than for those that did not. The result was the same for improvement in professional competence: Participants that preferred the methodology perceived an improvement in their competence. These results are similar to those obtained in other studies [31].

The training methodology is related to the applicability of the course contents to the workplace and the consequent improvement in professional competence [25].

These results evidence the interesting link among health care professionals’ satisfaction with training programs and the perceived improvements in their professional competence and health care quality. We may hypothesize that this training satisfaction is an important motivation indicator of their professional development. Learning by Doing strategies are prioritized by the sample, as a methodology which is useful in empowering professional performance. Another study with different cultural backgrounds showed that nurses in Mongolia felt professionally empowered by having gained informed evidence–based knowledge [44]. The auto efficacy in their own clinical competence could contribute to increasing job pride and satisfaction [44]. Changes in professional confidence and job satisfaction are related to improvements in the practice [45].

### Limitations and Future Work

A limitation of this present work would be the high level of collinearity among the independent variables. It would be a good idea to extend the evaluation process to include more cases and other methodologies, including qualitative techniques (discussion groups, field observations, in-depth interviews, etc.). A longitudinal study to examine the evolution of results through time is also recommended.

The Internet survey system requires alternative strategies of evaluation. In our research, 50% of potential subjects answered our questionnaire online. Health care professionals could face barriers to using the web due to access problems, time restrictions, and request of effort investment. Most of the questions in this format are devoted to measuring attitudes and perceptions, not behaviors [46]. A promoter of the research is the administration, and this instrument could be perceived as a tool for workplace control. Negative features have to be balanced with positive aspects, for example easy access to new and spread populations, cost savings, and anonymity [46].

It would be interesting to compare the professionals’ assessment of the quality of care with the opinions of the patients [47]. A study by Hernán, Gutiérrez, Lineros, Ruiz, and Rabadán [48] reported that the professionals indicated that there could be a relationship between their vision of quality of care and that of the patients. The investigation of both perceptions could provide interesting conclusions; it is something that requires further research and possible methodological triangulation [49].

From an institutional perspective, it would also be useful to undertake an extensive study of the good practices in training and evaluation that are developed by healthcare centers that are known to successfully motivate their professionals in the areas of occupational autonomy, organizational improvement, resource investment, evaluation, and innovation [50].

## 5. Conclusions

In this work, we developed a framework for assessing the relation between health care professionals’ modality of training and the evaluation of this experience by the health professionals enrolled in this training. The use of Learning by Doing strategy is valued by health care participants as one of the best teaching practices based on the workplace (health care settings). This satisfaction consensus has been taken into account not only during the training program but especially after it. Improvements in their skills, competences, and the quality of the care are results related to knowledge learning and transferring. These improvements are crucial for the health care professionals, organizations, and patients who could be benefit for them. Health policies related to health care professionals could consider Learning by Doing programs as a key resource to develop learning and improvement processes in their agents and organizations.

## Figures and Tables

**Figure 1 ijerph-16-01397-f001:**
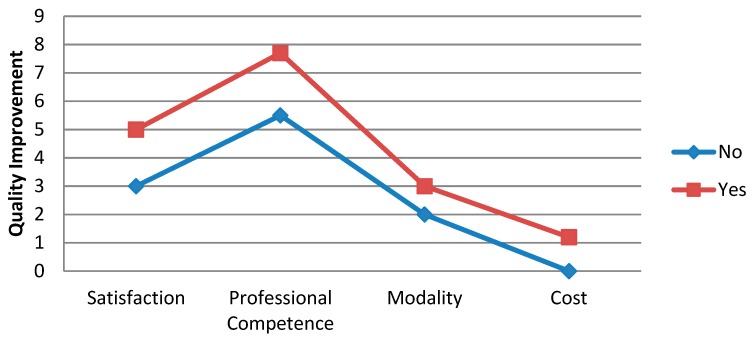
Comparison of the accumulated mean scores of the variables in relation to the improvement in the quality of health care.

**Table 1 ijerph-16-01397-t001:** Means comparison by dimensions and dependent variables.

Means Comparison	Applicability of Knowledge/Skills				Improvement of Knowledge/Skills i				Transmission of Knowledge/Skills			
Dimensions of Satisfaction	Likert	Response	Values	*p* Value	Likert	Response	Values	*p* Value	Likert	Response	Values	*p* Value
Course programme	0–5	No	4.09 (1.41)	0.013	0–5	No	3.64 (1.87)	0.005	0–5	No	2.83 (1.72)	0.000
		Yes	4.76 (0.64)			Yes	4.77 (0.55)			Yes	4.75 (0.66)	
Training methodology	0–4	No	2.25 (0.50)	0.154	0–4	No	2.67 (0.49)	0.003	0–4	No	2.25 (0.50)	0.003
		Yes	3.11 (0.54)			Yes	3.15 (0.52)			Yes	3.15 (0.54)	
Improvement of Professional Competence	0–7	No	5.65 (1.37)	0.000	0–7	No	4.69 (0.95)	0.000	0–8	No	5.50 (2.07)	0.007
		Yes	6.67 (1.05)			Yes	6.72 (1.05)			Yes	7.70 (1.12)	
Economic impact	0–3	No	0.53 (1.01)	0.001	0–2	No	0.20 (0.42)	0.000	0–2	No	0.14 (0.38)	0.001
		Yes	1.33 (1.10)			Yes	1.30 (1.11)			Yes	1.34 (1.12)	

**Table 2 ijerph-16-01397-t002:** Improvement of knowledge/skills and applicability by perceived improvement in quality of health care.

**Perceived Improvement in Quality of Health Care**	**Improvement of Knowledge/Skills**		**Total**
	**No**	**Yes**	
No	(4)	(3)	(7)
	28.60%	2.00%	4.30%
Yes	(10)	(146)	(156)
	71.40%	98%	95.70%
Total	(14)	(1.49)	(163)
	100%	100%	100%
**Perceived Improvement in Quality of Health Care**	**Applicability of Knowledge/Kills**		**Total**
	**No**	**Yes**	
No	(4)	(3)	(7)
	40.00%	2.00%	4.40%
Yes	(6)	(147)	(153)
	60.00%	98.00%	95.60%
Total	(10)	(150)	(160)
	100%	100%	100%

**Table 3 ijerph-16-01397-t003:** Crude and adjusted ORs by variable: Perceived improvement in the quality of health care.

Independent Variables	*p* Value	CRUDE OR (95% CI)	P	ADJUSTED OR (95% CI)
Satisfaction	0.001	3.125 (1.602–6.095)	0.329	1.772 (0.562–5.588)
Training methodology	0.006	25.802 (2.588–257.268)	0.034	16.479 (1.244–218.353)
Professional competence	0.001	2.999 (1.603–5.612)	0.874	0.916 (0.310–218.353)
Economic impact	0.019	12.744 (1.524–106.544)	0.161	7.174 (0.456–112.959)
Area Under Curve				0.930 (0–1)

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
