# Peer review of "Learning by Doing and Training Satisfaction: An Evaluation by Health Care Professionals"

_ijerph, 2019, doi:10.3390/ijerph16081397_

Reviewer 1 Report

I would like to congratulate the authors for their research. Just a minor comment: when you express the average results in each dimension, define the range, as you did in dimension 2 general satisfaction (0-5).

Author Response

 Comment: I would like to congratulate the authors for their research. Just a minor comment: when you express the average results in each dimension, define the range, as you did in dimension 2 general satisfaction (0-5)

Authors’ reply:

We thank a lot this comment as a source of motivation for both going on with our work and improving it.

We also have defined the ranges in pp 4-5.

1.        Satisfaction with the training methodology: (…) The average score (0 to 4) was 3.11 with a standard deviation of 0.54.

2.        Satisfaction with the training program (…) The average score for general satisfaction (0 to 5) was 4.67 with a standard deviation of 0.82.

3.        Satisfaction with the improvement in professional competence: (…) The average score (0 to 9) for the variables related to professional competence was 7.49 with a standard deviation of 1.31.

4.        Satisfaction with the improvement in quality of care: 95.7% said that they felt the course had improved their diagnostic, treatment and care skills at work (scores of 0 = no and 1 = yes).

5.        Satisfaction with the economic repercussions: (….) The average score (0 to 3) was 1.25 with a standard deviation of 1.11.

Please, take into account that a similar and more detailed explanation about variables and dimesions measures are provided in previous 2.3. Instrument section

Reviewer 2 Report

First, thank you for having the opportunity to review the manuscript "Learning by Doing and Training Satisfaction: An Evaluation by Health Care Professionals." Very interesting article for its publication, but it would require some improvements.

In the introduction, review lines 55-59.

Define the inclusion criteria of the participants.

There should be a section with data analysis. Clarify in this, the analysis performed, what statistical tests have been used? Why have non-paramedical tests been used?

It lacks a section with ethical considerations of the study.

The discussion is very superficial, there are hardly any studies with which to compare or discuss the results obtained. I advise to deepen more in the obtained findings.

Very old references, should be more updated.

Author Response

Comment: First, thank you for having the opportunity to review the manuscript "Learning by Doing and Training Satisfaction: An Evaluation by Health Care Professionals." Very interesting article for its publication, but it would require some improvements.

Authors’ reply: Thanks for your kind comment. The text has been amended in the light of the following comments.

Comment:  In the introduction, review lines 55-59.

Authors’ reply: We agree the need of the revision, and we have rewritten this paragraph: In the health care services, a promising research line has linked job satisfaction to quality of working life and work environment. Quality of Working Life (QWL) is a complex concept that contemplates: working hours, salary, the workplace environment, benefits and services, possibilities for professional promotion and improvements, access to information, human relationships and other factors that are relevant to job satisfaction and performance [9].

Comment: Define the inclusion criteria of the participants.

Authors’ reply: We have included the following comment: In this way, the 364 health care professionals who joined to the training program (Formative Focus 2014) were included in this evaluation process.

Please, take note that we have previously explained that the sample and the universe of this study is the same (Line: 136: 364 doctors and nurses participated. Given the number of subjects it was decided that it was not necessary to select a smaller sample, so the study universe became, in effect, the final sample)

Comment: There should be a section with data analysis. Clarify in this, the analysis performed, what statistical tests have been used? Why have non-paramedical tests been used?

Authors’ reply: We understand that chi-square is appropriate when the following conditions are met: As sampling method, we chose the simple random sampling. The variable under study is categorical. The expected value of the number of sample observations in each level of the variable is at least 5. We use the Chi-Square test to show the relationship between the variables and was replaced by Fisher´s exact text when these application criteria were not meet.

To compare means between independent groups, the Mann-Whitney U tests whether two samples are drawn from the same distribution, as compared to a given alternative hypothesis.

Finally, a predictive model for the variable related to quality of care was formulated using logistic regressions (Table 3) with unified values for four of the five dimensions. This test is not based on normality measurement level.

We did not include a section with data analysis because the different statistical procedures, were integrated in the sections 3.1. Descriptive results and 3.2. Inferential results. We did not wanted to be redundant with this information but of course, we could provide more details if reviewer considers it convenient, maybe in 2.1. Design section.

Comment: It lacks a section with ethical considerations of the study.

Authors’ reply: After conclusions we have included this new section about ethical considerations of the study:

Ethics approval and consent to participate

We obtained ethical approval from the Committee of Research Ethics of the Community of Aragon (CEICA) (Spain). This Committee is in charge of the evaluation of all research projects with people, human biological samples or personal data, including clinical trials and observational studies with medication. All study participants read the information sheet and agreed to participate in the study voluntarily. They answered the questions and consent by written.

Comment: The discussion is very superficial, there are hardly any studies with which to compare or discuss the results obtained. I advise to deepen more in the obtained findings.

Authors’ reply: The following paragraphs have been included along the discussion:

The statistical significance tests showed that learning is related to the applicability of the new knowledge and skills, the improvement of professional competence and the improvement in quality of care. However, this data is divergent with the results of other studies about health care environment where knowledge and attitude changes due to education programs, remain only for short period of time [37-39].

Therefore, follow up intervention programs and assessments linked to education and practice require to be planned as a key point of the continuous training [40].

In our research, special attention should be paid to the training methodology: it was unanimously agreed that Learning by Doing was the most suitable method for workplace training, all participants (100%) said that they preferred it to alternatives such as face-to-face classes, seminars, workshops or conferences. This preference could be explained by the education tradition in the health care professions as a form of apprenticeship [41] in which it is important to have direct and active experience, for example by interprofessional simulation programs [42]. In this sense, the program that we have evaluated (Formative Focus) combines the learning by doing strategy with interprofessional collaboration. In previous studies this combination has obtained excellent results also in terms of patient safety [43].

The bivariate analysis (…..) [31].

The training methodology is related to the applicability of the course contents to the workplace and the consequent improvement in professional competence [25].

These results evidence the interesting link among health care professional’s satisfaction with training programs and the perceived improvements in their professional competence and health care quality.

We may hypothesize that this training satisfaction is an important motivation indicator of their professional development. Learning by doing strategies are prioritized by the sample, as a methodology which is useful to empower professional performance. Other study with different cultural backgrounds shows that nurses in Mongolia feel professionally empowered by having gained informed evidence – based knowledge [44]. The auto efficacy in own clinical competence could contribute to increase the job proud and satisfaction [44]. Changes in professional confidence and job satisfaction are related to improvements in the practice [45].

Comment: Very old references, should be more updated.

Authors’ reply: The following updated references have been included:

1.        World Health Organization. Health Systems Governance for Universal Health Coverage Action Plan. WHO Publications: Geneva, Switzerland, 2014.

2.        Donabedian, A. Evaluating the quality of medical care. Milbank Q. 2005; 83, 691-729.

3.        Halldorsdottir, S.; Einarsdottir, E.J.; Edvardsson, I.R. Effects of cutbacks on motivating factors among nurses in primary health care. Scand J Caring Sci. 2018, 32, 397-406.

4.        Argimón, J.M., Jiménez, J. Métodos de investigación en clínica y epidemiología. Elsevier; Madrid, 2004.

5.        Wellings, C.A.; Gendek, M.A.; Gallagher, S.E. Evaluating Continuing Nursing Education. J Nur Prof Dev. 2017, 33, 281-286.

6.        Lapkin, S.; Levett-Jones, T.; Gilligan, C.A. A systematic review of the effectiveness of interprofessional education in health professional programs. Nurse Educ Today 2013, 33, 90-102.

7.        Wildeman, M.A.; Fles, R.; Adham, M.; Mayangsari, I.D.; Luirink, I. Short-term effect of different teaching methods on nasopharyngeal carcinoma for general practitioners in Jakarta, Indonesia. PloS One 2012, 3, 1-7.

8.        Moll, A.; Lambert, S.; Visker, J.; Dunseith, N.; Wang, A.; Azim, S. A case study activity to assess nursing students’ perceptions of their interprofessional healthcare team´s collaborative decision-making process. J Interprofessional Edu Prac. 2019, 14, 18-21.

9.        Sick, B.; Hager, K.D.; Uden, D.; Friederich, Ch.; Kim, H.M.; Branch-Mays, G.; Pittenger, A. Using a guided reflection tool and debriefing session to learn from interprofessional team interactions in clinical settings. J Interprofessional Edu Prac. 2019, 14, 53-57.

10.    Kilby, K.A.; Grajny, A.M.; Guarino, A.J.; Paniszyn, L.A.; McErlean, M. Experiental learning exercise to achieve objective assessment of interprofessional education. J Interprofessional Edu Prac. 2019, 14, 48-50.

11.    Hollamby, J.; Taylor, I.; Berragan, E.; Taylor, D.; Morgan, J. Preparing students for safe practice using an interprofessional ward simulation. J Interprofessional Edu Prac. 2019, 14, 78-82.

12.    Yoshino, Y.; Willott, C.; Gendenjamtz, E.; Surenkhorloo, A.; Islam, M.; Sakashita, R. Outcome Evaluation of Web Based Learning and Continuing Education Program for Maternal and Child Health Nursing and Other Professionals in Mongolia. Cen Asian J Med Sci. 2018, 4, 253-263.

13.    Tylre, S.; Bourbon, E.; Co, S.; Day, N.; Fineran, C.; Clinical Competency, Self-Efficacy, and Job Satisfaction Perceptions of the Staff Nurse. JNCD. 2012, 28, 32-35.

14.    Rice, S.; Winter, S.R.; Doherty S.; Milner, M. Advantages and Disadvantages of Using Internet-Based Survey Methods in Aviation-Related Research. JATE 2017, 7, 58–65.

Reviewer 3 Report

Abstract: Nothing to declare.

Introduction: Nothing to declare.

Material and Methods:

Line 149 - “The fact that the questionnaire was online may have influenced the number of replies.”: this aspect, could be explored in the discussion.

Line 148 - The validity of the instrument was tested in 10 health care professionals (5 physicians, 5 nursing staff). The reason for this sample size was not explained.

Results:
Table 1:  should be given a spacing between each of the “Dimensions of Satisfaction”, for easier reading of the table.
Line 247 - where it is written “Figure 1”, should be written “Graphic 1”.

Table 2: It should be displayed on the same page, for easier reading of the table.
The relative and absolute frequencies are presented, one of them should be placed in parentheses “(…)”, for a more easy reading of the table.

Table 3: What it means “AUC”, “area under the curve”?

Discussion:

Could be improved. The limitations represent more than half of the discussion.

Conclusions:

In several paragraphs (two) it is written "students", i think it would be "health professionals" (line 342, 347).

Author Response

Comment: Line 149 - “The fact that the questionnaire was online may have influenced the number of replies.”: this aspect, could be explored in the discussion.

Authors’ reply: We have explained this question in discussion section

The Internet survey system requires alternative strategies of evaluation. In our research, 50% of potential subjects answered our questionnaire on line. Health care professionals could have barriers to use the web due to access problems, time restrictions and request of effort investment. Most of the questions in this format are devoted to measure attitudes and perceptions, not behaviors [46]. Promoter of the research is the Administration and this instrument could be perceived as a tool for workplace control. Negative features have to be balanced with positive aspects, as for example: easy access to new and spread populations, cost savings and anonymity [46].

Comment: Line 148 - The validity of the instrument was tested in 10 health care professionals (5 physicians, 5 nursing staff). The reason for this sample size was not explained.

Authors’ reply: In order to explain in a proper way the size, we have added a new paragraph to this line.

To carry out the content validity of the questionnaire, the technique "expert opinion" was used. As recommended by the technique, it is advisable to consult a number of judges representative of the different groups in the sample and higher than the number of dimensions to be evaluated (from double to ten times according to Argimón and Jimenez, 36). Hence, to assess the final instrument of 5 dimensions (Satisfaction with the training methodology, Satisfaction with the training program, Satisfaction with the improvement in professional competence, Satisfaction with the improvement in quality of care, Satisfaction with the economic repercussions), a sample of 10 representative health professionals was selected (5 physicians, 5 nursing staff).

Comment: Table 1: should be given a spacing between each of the “Dimensions of Satisfaction”, for easier reading of the table.

Authors’ reply: Thanks for the suggestion. Please see new version of table 1 in the article

Comment: Line 247 - where it is written “Figure 1”, should be written “Graphic 1”.

Authors’ reply: Sorry for this mistake. It is already corrected.

Comment: Table 2: It should be displayed on the same page, for easier reading of the table.

Authors’ reply: In this new version Table 2 and Table 3 are not cut (p. 8). This comment and next one, allow us to improve the reading of the data. Thanks.

Comment: The relative and absolute frequencies are presented, one of them should be placed in parentheses “(…)”, for a more easy reading of the table.

Authors’ reply: Absolute frequencies are presented in parenthesis in Table 2 now

Comment: Table 3: What it means “AUC”, “area under the curve”?

Authors’ reply: Yes, we have rewritten the expression and correct a mistake in the same table.

Comment: Discussion: Could be improved. The limitations represent more than half of the discussion.

Authors’ reply:  The following paragraphs have been added to the text:

The statistical significance tests showed that learning is related to the applicability of the new knowledge and skills, the improvement of professional competence and the improvement in quality of care. However, this data is divergent with the results of other studies about health care environment where knowledge and attitude changes due to education programs, remain only for short period of time [37-39].

Therefore, follow up intervention programs and assessments linked to education and practice require to be planned as a key point of the continuous training [40].

In our research, special attention should be paid to the training methodology: it was unanimously agreed that Learning by Doing was the most suitable method for workplace training, all participants (100%) said that they preferred it to alternatives such as face-to-face classes, seminars, workshops or conferences. This preference could be explained by the education tradition in the health care professions as a form of apprenticeship [41] in which it is important to have direct and active experience, for example by interprofessional simulation programs [42]. In this sense, the program that we have evaluated (Formative Focus) combines the learning by doing strategy with interprofessional collaboration. In previous studies this combination has obtained excellent results also in terms of patient safety [43].

The bivariate analysis (…..) [31].

The training methodology is related to the applicability of the course contents to the workplace and the consequent improvement in professional competence [25].

These results evidence the interesting link among health care professional’s satisfaction with training programs and the perceived improvements in their professional competence and health care quality.

We may hypothesize that this training satisfaction is an important motivation indicator of their professional development. Learning by doing strategies are prioritized by the sample, as a methodology which is useful to empower professional performance. Other study with different cultural backgrounds shows that nurses in Mongolia feel professionally empowered by having gained informed evidence – based knowledge [44]. The auto efficacy in own clinical competence could contribute to increase the job proud and satisfaction [44]. Changes in professional confidence and job satisfaction are related to improvements in the practice [45].

Comment: Conclusions: In several paragraphs (two) it is written "students", i think it would be "health professionals" (line 342, 347).

Authors’ reply:  We appreciate the suggestion; this change has been done.

Round  2

Reviewer 2 Report

The changes are appropriate.

Author Response

Thank you very much for your valuable comments.